# Understanding the burden of poor mental health and wellbeing among persons affected by leprosy or Buruli ulcer in Nigeria: A community based cross-sectional study

Edmund Ndudi Ossai[1]*, Ngozi Ekeke[2], Amaka Esmai-Onyima[3], Chinwe Eze[2], Francis Chinawa[4], Obiora Iteke[4], Precious Henry[2], Joseph N. Chukwu[5], Charles Nwafor[2], Ngozi Murphy-Okpala[2], Martin Njoku[2], Anthony O. Meka[2], Chukwuma Anyaike[6]

1 Department of Community Medicine, College of Health Sciences, Ebonyi State University, Abakaliki, Nigeria, 2 Red Aid Nigeria (RAN) Enugu, Enugu, Nigeria, 3 Tuberculosis, Leprosy and Buruli ulcer Control Unit, Ogbaru, Anambra State, Nigeria, 4 Federal Neuro-Psychiatric Hospital, Enugu, Nigeria, 5 German Leprosy and TB Relief Association Enugu, Enugu, Nigeria, 6 National Tuberculosis, Leprosy and Buruli ulcer Control Programme (NTBLCP), Abuja, Nigeria

* ossai_2@yahoo.co.uk

**Data Availability Statement:** All relevant data are within the manuscript and its supporting information files.

## Abstract

### Background

Skin neglected tropical diseases including leprosy and Buruli ulcer (BU)are a group of stigmatizing and disability-inducing conditions and these aspects of the diseases could lead to poor mental health. The study was designed to assess the burden of poor mental health and wellbeing among persons affected by leprosy or BU in Nigeria

### Methods

A community based cross-sectional study design was employed. The study involved persons affected by leprosy or BU. Ten local government areas with the highest number of notified leprosy or BU cases between 2014 and 2018 in southern Nigeria were purposively selected. Information were obtained using Patient Health Questionnaire-9 (PHQ-9), Generalized Anxiety Disorders-7 (GAD-7), Warwick-Edinburgh Mental Well-being Scale (WEMWBS) and OSLO Social Support Scale. Outcome measure was poor mental health/ wellbeing and was determined by proportion of respondents who had depressive symptoms, anxiety disorder and poor mental wellbeing.

### Results

A total of 635 persons affected by leprosy or BU participated in the study. The mean age of respondents was 43.8±17.0 years and highest proportion, 22.2% were in age group, 40–49 years. Majority of respondents, 50.7% were males. A higher proportion of respondents, 89.9% had depressive symptoms, 79.4% had anxiety disorders and 66.1% had poor mental wellbeing. Majority, 57.2% had poor mental health/wellbeing. Among the respondents, there was a strong positive correlation between depression and anxiety scores, (r = 0.772,

**Funding:** NE, ENO and AE received the grant. (Reference Number, 708.20.15/LRI). Leprosy Research Initiative (LRI) https://leprosyresearch.org/ The funders played no role in the design of the study, data collection and analysis, decision to publish and in the preparation of the manuscript.

**Competing interests:** The authors have declared that no competing interests exist.

**Abbreviations:** AOR, Adjusted odds ratio; BU, Buruli ulcer; CI, Confidence interval; FCT, Federal capital territory; GAD, Generalized anxiety disorder; LGA, Local government area; NTBLCP, National tuberculosis, leprosy and Buruli ulcer control program; NTD, Neglected tropical disease; PHQ, Patient health questionnaire; SPSS, Statistical Product for Service Solutions; TB, Tuberculosis; WEMWBS, Warwick-Edinburgh mental well-being scale; WHO, World Health Organization.

p<0.001). There was a weak negative correlation between depression score and WEMWBS score, (r = -0.457, p<0.001); anxiety score and WEMWBS score, (r = -0.483, p<0.001). Predictors of poor mental health/wellbeing included having no formal education, (AOR = 1.9, 95%CI: 1.1–3.3), being unemployed, (AOR = 3.4, 95%CI: 2.2–5.3), being affected by leprosy, (AOR = 0.2, 95%CI: 0.1–0.4) and having poor social support, (AOR = 6.6, 95%CI: 3.7–11.8).

## Conclusions

The burden of poor mental health/wellbeing among persons affected by leprosy or BU is very high. There is need to include mental health interventions in the management of persons affected with leprosy or BU. Equally important is finding a feasible, cost-effective and sustainable approach to delivering mental health care for persons affected with leprosy or BU at the community level. Improving educational status and social support of persons affected by leprosy or BU are essential. Engaging them in productive activities will be of essence.

## Introduction

Neglected tropical diseases including leprosy and Buruli ulcer (BU), present with disruptions of the skin barrier that are associated with scarring. Both are chronic diseases and while leprosy is caused by a type of bacteria named mycobacterium leprae, BU is caused by mycobacterium ulcerans. The changes associated with the diseases burden the lives of those affected through reduced mobility, hindered productivity and psychological distress [1, 2]. Thus, they are disability inducing infections. In recent times, there has been a focus on NTDs because they are concentrated among the poorest of the world [3]. Moreover, exclusion due to leprosy because of physical and sensory impairments may worsen the mental health of persons affected by the disease [4]. There is evidence also that leprosy is associated with stigma and stigma is linked with both exclusion and poor mental health [5, 6].

A study in Ghana affirmed that BU disease is associated with mental health distress among persons affected by the disease and also their caregivers [7]. Thus the researchers concluded on the need for integration of psychosocial interventions in the management of the disease [7]. Before then, persons affected by BU have echoed the need for inclusion of psychological care in their health care needs [8]. Subsequently, a study in Nepal revealed that the prevalence of depression among persons affected by leprosy was about five times higher than that of community members [9]. Similarly, the risk of developing psychiatric morbidity was found to be significantly higher among persons affected with leprosy than those with albinism [10]. There has been a postulation that NTDs are major drivers of mental ill health in persons affected by the diseases, their families and caregivers [6].Thus, persons affected with NTDs are at a higher risk of mental health conditions and individuals with mental health illnesses are also at a higher risk of being affected by an NTD [11].

During the COVID-19 pandemic, persons affected by leprosy had deep personal experiences with some of the consequences of the pandemic. These included quarantining, isolation, loneliness, lost livelihoods and fear of transmission of the disease. Suffice it to say that some of them perceived the trauma of the COVID-19 period as being similar to their experience of being isolated because of leprosy [12]. Thus, the COVID-19 pandemic adversely affected the

mental health of persons affected by leprosy. This was collaborated by the finding of a study in India where it was found that fifty percent of persons affected by leprosy reported worsening mental health conditions as a result of the pandemic [13]. Also, a recent systematic review revealed that as a consequence of their mental health, low self-esteem and low quality of life were found among those affected by leprosy and their children [14]. These findings have necessitated the call for the integration of evidence based mental health care into NTD programs worldwide [6]. Moreover, a study in northwest Nigeria emphasized the need for a holistic approach in the management of persons affected by leprosy with specific attention to the mental health needs of such individuals [15]. Furthermore, in the year 2019, the World Health Organization launched the WHO Special Initiative for Mental Health tagged Universal Health Coverage for Mental Health. The aim of the initiative was to ensure access to quality and affordable care for mental health conditions in twelve priority countries which is expected to reach 100 million more people [16]. This study was therefore designed to assess the mental health and wellbeing of persons affected by leprosy or BU in Nigeria.

## Methodology

### Study setting

Nigeria is the most populous country in Africa and the current population is estimated to be 227,236,177 by February 2024 [17]. The country is divided into 36 States and a Federal Capital Territory (FCT). The states are further classified into six geo-political zones. Each state is divided into Local Government Areas (LGAs), which represent the third tier of government. There are a total of 774 LGAs. The National Tuberculosis and Leprosy Control Program (NTBLCP) under the Federal Ministry of Health was inaugurated in 1988. The NTBLCP was initially responsible for tuberculosis and leprosy control but later included BU to its mandate. Nigeria despite achieving the WHO leprosy elimination target of less than 1 per 10,000 population in 1998 continues to have pockets of high endemicity of the disease in some States and LGAs across the country. The study was conducted in two geo-political zones in southern Nigeria including Anambra and Ebonyi states in southeast Nigeria and Delta, Akwa-Ibom, Cross River and Bayelsa states in south-south geo-political zone. The ten LGAs selected were Ogoja, Calabar South and Obubra LGAs, (Cross River State) Ogbaru and Anambra East LGAs (Anambra state). Others included Ogbia, (Bayelsa state); Etinan, (Akwa-Ibom state); Ebonyi LGA, (Ebonyi state) and Ethiope East and Isoko South LGAs in Delta state.

### Study design

This was a community based cross-sectional study. Ten LGAs in six states of Southern Nigeria with the highest number of notified leprosy or BU cases between 2014 and 2018 were purposively selected from 220 LGAs in south-east and south-south geo-political zones of Nigeria. The study took place between February and June 2021.

### Study population

This included persons affected by leprosy or BU who have been registered for treatment in the ten selected LGAs under the care of NTBLCP who were willing to participate in the study. For persons affected by leprosy only those who were 18 years and above were included in the study. However, because of the epidemiology of BU in the country, age was not a barrier for inclusion of persons affected by BU in the study. All persons affected by leprosy and BU who failed to give consent to participate and those who were very ill during the period of the study were excluded.

## Sampling technique

This was a total population study of all persons affected by leprosy or BU in the selected local government areas. There are no official records of number of persons affected by leprosy or BU in the selected LGAs but efforts were made to include all persons that met the inclusion criteria into the study through the help of the officers of the various State TB Leprosy and BU Control Programs and other persons affected by either of the two diseases. A total of 635 persons affected by leprosy or BU were included in the study.

## Data collection tools/methods

In assessing the mental health and wellbeing of persons affected by leprosy and BU, the following tools were used; Patient Health Questionnaire-9 (PHQ-9); Generalized Anxiety Disorders-7 (GAD-7) and Warwick-Edinburgh Mental Well-being Scale (WEMWBS) among eligible persons affected by leprosy or BU. The OSLO Social support scale was used to assess social support among the respondents. The questionnaire for the study also included a section on socio-demographic characteristics.

## Patient Health Questionnaire 9 (PHQ-9)

This is a validated questionnaire that is used for screening, diagnosing, monitoring and measuring the severity of depressive symptoms [18]. It is made up of nine variables and responses are in a Likert scale format with scores ranging from zero to three. Thus the minimum score that could be obtained for each respondent was zero while the maximum score could be 27. The assessment of each of the variables was made with reference to the past two weeks. A score of zero meant Not at all; one meant over several days; two denoted more than half the days and three meant almost every day. The PHQ 9 score was obtained by adding the score for each of the nine variables. The total PHQ score was then categorized as follows

    Score of zero to 4 No depression
    Score of 5 to 9 Mild depression
    Score of 10 to 14 Moderate depression
    Score of 15 to 19 Moderately severe depression
    Score of 20 to 27 Severe depression
    The PHQ 9 score was further categorized as follows, a score of zero to 4 indicated No
depression while a score of 5 to 27 meant there was depression.

## The Generalized Anxiety Disorder 7 (GAD 7)

This is a validated questionnaire that is used as an initial screening tool for generalized anxiety disorder [19]. It is made up of seven variables and responses are in a Likert scale format with scores ranging from zero to three. Thus the minimum score that could be obtained for each respondent was zero while the maximum score could be 21. The assessment of each of the variables was made with reference to experiences in the past two weeks. A score of zero meant Not at all; one meant over several days; two denoted more than half the days and three meant almost every day. The total GAD 7 score was obtained by adding the score for each of the seven variables. The total GAD score was then categorized as follows

    Score of zero to 4 No anxiety
    Score of 5 to 9 Mild anxiety
    Score of 10 to 14 Moderate anxiety
    Score of more than 15 Severe anxiety

The total GAD 7 score was further categorized as follows, a score of zero to 4 indicated no anxiety and a score of 5 and above meant there was anxiety disorder.

### The Warwick-Edinburgh Mental Wellbeing Scale (WEMWBS)

The Warwick-Edinburgh Mental Well-being Scale (WEMWBS) is a validated questionnaire made up of 14 positively worded items for assessing the mental well-being of a population [20]. The questionnaire was developed in 2006 through the combined efforts of Warwick and Edinburgh universities in a bid to develop the Scottish mental health indicators for adults. There is also a shortened version of the scale which is made up of seven variables. The items in the questionnaire cover both the feeling and functional aspects of mental wellbeing. Each of the 14-item scale has five response categories ranging from one to five which is summed up to provide a single score. The higher the total score, the higher the mental wellbeing of the respondent. The least score that could be attained by any respondent(was 14 while the highest score could be 70. The total WEMWBS score for each respondents was categorized into two, good and poor mental wellbeing using a mean WEMWBS score of 49.4 [21]. Respondents that had total WEMWBS scores of 49.4 and below were categorized as having poor mental wellbeing.

Data was collected from persons affected by leprosy or BU by trained research assistants. Efforts were made to reach the persons affected by leprosy or BU in their various homes where they were interviewed. Assistances were obtained from the State TB, Leprosy and BU Control Program officers and also persons affected by leprosy or BU.

### Data analysis

Data entry and analysis were done using IBM Statistical Product for Service Solution (SPSS) statistical software version 25. Continuous variables were summarized using mean and standard deviation while categorical variables were presented using frequencies and proportions. Chi square test was used to determine the difference in proportions between two categorical variables while correlation analysis was used to determine the strength of linear relationship between two continuous variables. The level of statistical significance was determined by a p value of $<0.05$.

The outcome measure of the study was poor mental health/wellbeing among the respondents. This was determined by the proportion of the respondents who had depressive symptoms, anxiety disorder and also poor mental wellbeing. This was defined as poor mental health/wellbeing. Multivariate analysis using binary logistic regression was used to determine the predictors of poor mental health/wellbeing among the respondents. First the outcome variable was cross-tabulated with the socio-demographic characteristics of the respondents and other variables that follow a logical sequence. Variables that had a p value of $<0.2$ at bivariate analysis were entered into the logistic regression model to determine the predictors of poor mental health/wellbeing among the respondents. The results of the binary logistic regression analysis were presented using adjusted odds ratio and 95% confidence interval and the level of statistical significance was determined by a p value of $<0.05$.

### Ethics approval

Ethical approval for the study was obtained from the Health Research and Ethics Committee of University of Nigeria Teaching Hospital, Ituku Ozalla, Enugu.(NHREC 0501-2008B). The study took place between February and June 2021. The respondents signed or thumb printed to a written informed consent form before participation in the study and the extent of their involvement in the study was made known to them. For respondents who were less than 18

years of age, the parents/guardians of the respondents signed a written informed consent form while a verbal assent was obtained from the respondents which was witnessed by the parent/guardian. Participation in the study was voluntary. Also, respondents were assured that, there would be no victimization of respondents who refused to participate or who decided to withdraw from the study after giving consent. Respondents were assured that all information provided through the questionnaire will be kept confidential.

## Results

Table 1 shows the socio-demographic characteristics of the respondents. The mean age of the respondents was 43.8±17.0 years. The highest proportion of the respondents, 22.2% were in the age group 40–49 years while the least proportion, 17.6% were between 50 and 59 years. A higher proportion of the respondents, 50.7% were males. Majority of the respondents, 78.9% were affected by leprosy.

Table 2 shows the assessment of depressive symptoms among the respondents. The highest proportion of respondents, 31.8% had little interest or pleasure in doing things nearly every day while the least proportion, 17.3% did not have little interest or pleasure in doing things at all. The highest proportion of respondents, 34.0% feel tired or having little energy on several days while the least proportion, 12.9% did not have feel tired at all.

Table 3 shows the assessment of anxiety symptoms among the respondents. The highest proportion of the respondents, 30.9% feel nervous, anxious or on edge on several days while the least proportion, 20.9% did not feel nervous at all. The highest proportion of respondents, 35.3% had trouble relaxing on several days while the least proportion, 15.0% had trouble relaxing nearly every day.

Table 4 shows the assessment of mental well-being among the respondents. The highest proportion of the respondents, 27.6% have been feeling useful some of the time while the least proportion, 6.9% do not feel useful at all. The highest proportion of the respondents, 25.8% have been feeling good about themselves some of the time while the least proportion, 18.0% rarely feel good about themselves. The highest proportion of the respondents, 28.7% felt loved some of the time while the least proportion 10.7% do not feel loved at all.

Table 5 shows the prevalence of depressive and anxiety symptoms among the respondents. Majority of the respondents, 89.8% had depressive symptoms. Similarly, a higher proportion of the respondents, 79.4% had anxiety symptoms. Also, a higher proportion of the respondents 77.5% had both depressive and anxiety symptoms. A minor proportion of the respondents, 33.9% have good mental well-being.

Table 6 shows the comparison of depressive and anxiety symptoms based on gender. Comparable proportions of male, 77.6% and female respondents, 77.3% have both depressive and anxiety symptoms combined, ($\chi^2$ = 0.010, p = 0.922). A higher proportion of male respondents, 39.1% had good mental well-being when compared with the females, 28.4% and the difference in proportions was found to be statistically significant, ($\chi^2$ = 8.108, p = 0.004).

Table 7 shows the correlation matrix of variables. There was a strong positive correlation between depression score and anxiety score of respondents, increases in depression correlates with increases in anxiety and this was found to be statistically significant, (n = 635, r = 0.772, p<0.001). There was a weak negative correlation between anxiety score and WEMWBS score, increases in anxiety score correlates with decreases in WEMWBS score and this was found to be statistically significant, (n = 635, r = -0.483, p<001).

Table 8 shows the factors affecting Poor mental health/wellbeing among the respondents. The respondents who had no formal education were about twice more likely to have poor mental health/wellbeing when compared with those who had secondary education and above,

**Table 1. Socio-demographic characteristics of persons affected by leprosy or Buruli ulcer in Nigeria (n = 635).**

| Variable | Frequency | Percent (%) |
|---|---|---|
| **Age of respondents** | | |
| Mean±(SD) | 43.8±17.0 | |
| **Age of respondents in groups** | | |
| <30 years | 127 | 20.0 |
| 30–39 years | 122 | 19.2 |
| 40–49 years | 141 | 22.2 |
| 50–59 years | 112 | 17.6 |
| ≥60 years | 133 | 20.9 |
| **Gender** | | |
| Male | 322 | 50.7 |
| Female | 313 | 49.3 |
| **Marital status** | | |
| Never married | 196 | 30.9 |
| Married | 341 | 53.7 |
| Separated/Divorced | 44 | 6.9 |
| Widowed | 54 | 8.5 |
| **Religion** | | |
| Christianity | 607 | 95.6 |
| Islam | 1 | 0.2 |
| Traditional religion | 27 | 4.3 |
| **Educational attainment of respondents** | | |
| No formal education | 179 | 28.2 |
| Primary education | 268 | 42.2 |
| Secondary education | 160 | 25.2 |
| Tertiary education | 28 | 4.4 |
| **Employment status of respondent** | | |
| Unemployed | 242 | 38.1 |
| Self-employed | 369 | 58.1 |
| Paid employment | 24 | 3.8 |
| **Have had vocational training** | | |
| Yes | 57 | 9.0 |
| No | 578 | 91.0 |
| **Number of dependents** | | |
| None | 177 | 27.9 |
| 1–4 individuals | 203 | 32.0 |
| ≥5 individuals | 255 | 40.2 |
| **Diagnosis** | | |
| Leprosy | 501 | 78.9 |
| Buruli ulcer | 134 | 21.1 |

(AOR = 1.9, 95%CI: 1.1–3.3). Similarly, the respondents that had primary education were about twice more likely to have poor mental health/wellbeing when compared with those who had secondary education and above, (AOR = 1.7, 95%CI: 1.1–2.7). The respondents who were unemployed were 3.4 times more likely to have poor mental health/wellbeing when compared with those who were employed, (AOR = 3.4, 95%CI: 2.2–5.3). The respondents who were affected by leprosy were five times less likely to have poor mental health/wellbeing when compared with those who were diagnosed with BU. (AOR = 0.2, 95%CI: 0.1–0.4). The respondents

Table 2. Assessment of depressive symptoms among persons (n = 635) affected by leprosy or Buruli ulcer in Nigeria using PHQ9.

| Variable | Not at all N (%) | Several days N (%) | More than half the days N (%) | Nearly every day N (%) |
|---|---|---|---|---|
| Little interest or pleasure in doing things | 110 (17.3) | 184 (29.0) | 139 (21.9) | 202 (31.8) |
| Feeling down, depressed or hopeless | 120 (18.9) | 187 (29.4) | 150 (23.6) | 178 (28.0) |
| Trouble feeling or staying asleep or sleeping too much | 161 (25.4) | 168 (26.5) | 165 (26.0) | 141 (22.2) |
| Feeling tired or having little energy | 82 (12.9) | 216 (34.0) | 183 (28.8) | 154 (24.3) |
| Poor appetite or over eating | 199 (31.3) | 203 (32.0) | 137 (21.6) | 96 (15.1) |
| Feeling bad about oneself | 144 (22.7) | 205 (32.3) | 110 (17.3) | 176 (27.7) |
| Trouble concentrating on things | 158 (24.9) | 188 (29.6) | 152 (23.9) | 137 (21.6) |
| Moving or speaking so slowly or being fidgety or restless | 208 (32.8) | 179 (28.2) | 141 (22.2) | 107 (16.9) |
| Thoughts that you will be better off dead | 277 (43.6) | 152 (23.9) | 106 (16.7) | 100 (15.7) |

Table 3. Assessment of anxiety symptoms among persons (n = 635) affected by leprosy or Buruli ulcer in Nigeria using GAD7.

| Variable | Not at all N (%) | Several days N (%) | More than half the days N (%) | Nearly every day N (%) |
|---|---|---|---|---|
| Feeling nervous, anxious or on edge | 133 (20.9) | 196 (30.9) | 160 (25.2) | 146 (23.0) |
| Not being able to stop or control worrying | 125 (19.7) | 202 (31.8) | 151 (23.8) | 157 (24.7) |
| Worrying too much about different things | 125 (19.7) | 186 (29.3) | 155 (24.4) | 169 (26.6) |
| Trouble relaxing | 176 (27.7) | 224 (35.3) | 140 (22.0) | 95 (15.0) |
| Being so restless that it's hard to sit still | 227 (35.7) | 165 (26.0) | 136 (21.4) | 107 (16.9) |
| Becoming easily annoyed or irritable | 183 (28.8) | 176 (27.7) | 135 (21.3) | 141 (22.2) |
| Feeling afraid as if something awful might happen | 231 (36.4) | 151 (23.8) | 114 (18.0) | 139 (21.9) |

Table 4. Assessment of mental well-being among persons (n = 635) affected by leprosy or Buruli ulcer in Nigeria using WEMWBS.

| Variable | None of the time N (%) | Rarely N (%) | Some of the time N (%) | Often N (%) | All the same N (%) |
|---|---|---|---|---|---|
| I have been feeling optimistic about the future | 87 (13.7) | 106 (16.7) | 174 (27.4) | 108 (17.0) | 160 (25.2) |
| I have been feeling useful | 44 (6.9) | 121 (19.1) | 175 (27.6) | 144 (22.7) | 151 (23.8) |
| I have been feeling relaxed | 111 (17.5) | 135 (21.3) | 212 (33.4) | 83 (13.1) | 94 (14.8) |
| I have been feeling interested in other people | 82 (12.9) | 110 (17.3) | 190 (29.9) | 94 (14.8) | 159 (25.0) |
| I have had energy to spare | 99 (15.6) | 158 (24.9) | 171 (26.9) | 126 (19.8) | 81 (12.8) |
| I have been dealing with problems well | 98 (15.4) | 140 (22.0) | 224 (35.3) | 89 (14.0) | 84 (13.2) |
| I have been thinking clearly | 68 (10.7) | 128 (20.2) | 148 (23.3) | 139 (21.9) | 152 (23.9) |
| I have been feeling good about myself | 115 (18.1) | 114 (18.0) | 164 (25.8) | 115 (18.1) | 127 (20.0) |
| I have been feeling close to other people | 58 (9.1) | 127 (20.0) | 198 (31.2) | 123 (19.4) | 129 (20.3) |
| I have been feeling confident | 85 (13.4) | 117 (18.4) | 193 (30.4) | 113 (17.8) | 127 (20.0) |
| I have been able to make up my own minds about things | 45 (7.1) | 107 (16.9) | 195 (30.7) | 134 (21.1) | 154 (24.3) |
| I have been feeling loved | 68 (10.7) | 122 (19.2) | 182 (28.7) | 129 (20.3) | 134 (21.1) |
| I have been interested in new things | 148 (23.3) | 77 (12.1) | 149 (23.5) | 113 (17.8) | 148 (23.3) |
| I have been feeling cheerful | 35 (5.5) | 118 (18.6) | 279 (43.9) | 77 (12.1) | 126 (19.8) |

who had poor social support were about seven times more likely to have poor mental health/wellbeing when compared with those who had strong social support. (AOR = 6.6, 95%CI: 3.7–11.8). Similarly, the respondents who had moderate social support were three times more likely to have poor mental health/wellbeing when compared with those who had strong social support, (AOR = 3.0, 95%CI: 1.6–5.5).

**Table 5. Prevalence of depressive and anxiety symptoms among persons (n = 635) affected by leprosy or Buruli ulcer in Nigeria.**

| Variable | Frequency | Percent (%) |
|---|---|---|
| **Depression (categorized)** | | |
| No depression | 65 | 10.2 |
| Mild depression | 130 | 20.5 |
| Moderate depression | 190 | 29.9 |
| Moderately severe depression | 170 | 26.8 |
| Severe depression | 80 | 12.6 |
| **Depressive symptom** | | |
| Yes | 570 | 89.8 |
| No | 65 | 10.2 |
| **Anxiety (Categorized)** | | |
| No anxiety | 131 | 20.6 |
| Mild anxiety | 178 | 28.0 |
| Moderate anxiety | 183 | 28.8 |
| Severe anxiety | 143 | 22.5 |
| **Anxiety disorder** | | |
| Yes | 504 | 79.4 |
| No | 131 | 20.6 |
| **Depression and Anxiety combined** | | |
| Yes | 492 | 77.5 |
| No | 143 | 22.5 |
| **Mental well-being** | | |
| Poor | 420 | 66.1 |
| Good | 215 | 33.9 |
| **Mental health/Wellbeing** | | |
| Poor | 363 | 57.2 |
| Good | 272 | 42.8 |
| **Social support** | | |
| Poor support | 365 | 57.5 |
| Moderate support | 178 | 28.0 |
| Strong support | 92 | 14.5 |

## Discussion

From the results of this study, a very high proportion of respondents in the study, 89.8%% were found to have depressive disorders. This proportion of respondents who were depressed from the result of this study is higher than what was obtained from a study in Sokoto, Nigeria where 47.7% of the people with leprosy had psychiatric disorders [15]. Studies in India have indicated the range of depressive disorders from 30–53% for persons affected by leprosy [22–25]. The study also revealed that very high proportion of respondents 79.4% were found to have anxiety disorders. In a study in India that involved persons affected by leprosy who presented in the out-patient department of a hospital, 10% of them had anxiety disorders [23]. In a study that included persons affected by leprosy and lymphatic filariasis, about half of them reported episodes of depression and anxiety [24]. In another study in India, 19% of persons affected by leprosy were found to have anxiety disorders [22].

The most comparable result to that obtained in this study was obtained from a study in Bangladesh in which 97% of the respondents had depressive symptoms while the whole respondents had anxiety symptoms [26]. Among respondents in this study, 39.4% had

Table 6. Comparison of depressive and anxiety symptoms based on gender.

| Variable | Gender (n = 635) | | χ² | p value |
|---|---|---|---|---|
| | Male N (%) | Female N (%) | | |
| **Depressive symptom** | | | | |
| Yes | 287 (89.1) | 283 (90.4) | 0.285 | 0.593 |
| No | 35 (10.9) | 30 (9.6) | | |
| **Anxiety disorder** | | | | |
| Yes | 256 (79.5) | 248 (79.2) | 0.007 | 0.933 |
| No | 66 (20.5) | 65 (20.8) | | |
| **Depression and Anxiety combined** | | | | |
| Yes | 250 (77.6) | 242 (77.3) | 0.010 | 0.922 |
| No | 72 (22.4) | 71 (22.7) | | |
| **Mental wellbeing** | | | | |
| Poor | 196 (60.9) | 224 (71.6) | 8.108 | 0.004 |
| Good | 126 (39.1) | 89 (28.4) | | |
| **Mental health/Well-being** | | | | |
| Poor | 173 (53.7) | 190 (60.7) | 3.155 | 0.076 |
| Good | 149 (46.3) | 123 (39.3) | | |
| **Social support** | | | | |
| Poor support | 184 (57.1) | 181 (57.8) | 0.094 | 0.954 |
| Moderate support | 90 (28.0) | 88 (28.1) | | |
| Strong support | 48 (14.9) | 44 (14.1) | | |

moderate to severe depression while 20.5% had mild depression. From the results of the study in Bangladesh, 53% of respondents experienced moderate to severe depression while 44% had mild depression [26]. Similarly, among respondents in this study, 22.5% of the respondents had severe anxiety disorder. In the study in Bangladesh, 37% of the respondents had severe anxiety disorder [26]. These findings make it imperative that mental health interventions should be part of the management of persons affected by leprosy or Buruli ulcer.

Very high proportions of respondents in this study had depressive and anxiety disorders. It has been posited that persons affected with NTDs are at a higher risk of mental health conditions. [11]. It could be explained that the poor mental health state of persons affected by leprosy or BU could have been worsened by the COVID-19 pandemic as this study took place almost immediately after the lockdown period associated with the COVID-19 pandemic. For

Table 7. Correlation matrix of variables.

| | Correlation co-efficient r, p value, (n = 635) | | | | |
|---|---|---|---|---|---|
| | Age in years | PHQ 9 score | GAD 7 score | WEMWBS Score | Social support Scale |
| **Age in years** | 1 | | | | |
| **PHQ 9 score** | r = 0.092 | | | | |
| | p = 0.020 | 1 | | | |
| **GAD 7 Score** | r = 0.068 | r = 0.772 | | | |
| | p = 0.089 | p<0.001 | 1 | | |
| **WEMWBS score** | r = -0.001 | r = -0.457 | r = -0.483 | | |
| | p = 0.976 | p<0.001 | p<0.001 | 1 | |
| **Social Support Scale** | r = 0.006 | r = -0.353 | r = -0.463 | r = 0.464 | |
| | p = 0.873 | p<0.001 | p<0.001 | p<0.001 | 1 |

**Table 8. Factors affecting poor mental health/wellbeing among the respondents.**

| Variable | Mental Health/Wellbeing (n = 635) | | p value** | AOR(95%CI)*** |
|---|---|---|---|---|
| | Poor N (%) | Good N (%) | | |
| **Age of respondents in groups** | | | | |
| <40 years | 149 (59.8) | 100 (40.2) | 0.170 | 0.7 (0.4–1.2) |
| 40–49 years | 71 (50.4) | 70 (49.6) | | 0.7 (0.4–1.1) |
| ≥50 years | 143 (58.6) | 102 (41.6) | | 1 |
| **Gender** | | | | |
| Male | 173 (53.7) | 149 (46.3) | 0.076 | 0.9 (0.6–1.4) |
| Female | 190 (60.7) | 123 (39.3) | | 1 |
| **Marital status** | | | | |
| Married | 178 (52.2) | 163 (47.8) | 0.006 | 0.9 (0.6–1.4) |
| Single* | 185 (62.9) | 109 (37.1) | | 1 |
| **Educational attainment of respondents** | | | | |
| No formal education | 111 (62.0) | 68 (38.0) | 0.096 | 1.9 (1.1–3.3) |
| Primary education | 156 (58.2) | 112 (41.8) | | 1.7 (1.1–2.7) |
| Secondary education | 96 (51.1) | 92 (48.9) | | 1 |
| **Employment status of respondent** | | | | |
| Unemployed | 191 (78.9) | 51 (21.1) | <0.001 | 3.4 (2.2–5.3) |
| Employed | 172 (43.8) | 221 (56.2) | | 1 |
| **Have had vocational training** | | | | |
| Yes | 24 (42.1) | 33 (57.9) | 0.016 | 0.8 (0.4–1.5) |
| No | 339 (58.7) | 239 (41.3) | | 1 |
| **Number of dependents** | | | | |
| None | 123 (69.5) | 54 (30.5) | <0.001 | 0.9 (0.5–1.8) |
| 1–4 individuals | 102 (50.2) | 101 (49.8) | | 0.9 (0.6–1.4) |
| ≥5 individuals | 138 (54.1) | 117 (45.9) | | 1 |
| **Diagnosis of respondent** | | | | |
| Leprosy | 256 (51.1) | 245 (48.9) | <0.001 | 0.2 (0.1–0.4) |
| Buruli ulcer | 107 (79.9) | 27 (20.1) | | 1 |
| **Social support** | | | | |
| Poor social support | 261 (71.5) | 104 (28.5) | <0.001 | 6.6 (3.7–11.8) |
| Moderate social support | 82 (46.1) | 96 (53.9) | | 3.0 (1.6–5.5) |
| Strong social support | 20 (21.7) | 72 (78.3) | | 1 |

*Never married, widowed, divorced

**p value at bivariate analysis

***Adjusted odds ratio, 95% confidence interval

instance, the Young minds January 2021 study on the impact of COVID-19 on young people with mental health needs attested that 67% of young people believed that the COVID-19 pandemic would have a long-term negative effect on their mental health [27]. Perhaps, it could be said that the COVID-19 pandemic worsened the mental health of other groups of people and this included persons affected by leprosy or BU.

Before the COVID-19 pandemic, a study in Cuiaba, Brazil revealed that 70.4% of persons affected with leprosy had common mental disorders [28]. There is evidence that mental health conditions are higher among persons affected by leprosy when compared with other community members who are free of the disease [9] and also among those who had other skin conditions like albinism [10]. Similarly, a study in southern Nepal revealed that persons affected by

leprosy who belonged to self-help groups have significantly lower level of mental well-being and higher level of depression than that in the general population [29].

The proportion of respondents with poor mental well-being in this study was 66.1%. This finding was also higher than that obtained from a study in Nepal involving persons affected with leprosy who belonged to a self-help group. In that study, the proportion of respondents with poor mental well-being were 38% [29]. It could be said that belonging to a self-help group may have moderated the mental well-being of the respondents in the Nepal study thus comparatively improving their mental health when compared with respondents in this study.

From the results of this study, comparable proportions of male and female respondents had depressive or anxiety symptoms or both symptoms combined. In a study among medical students in a university in southeast Nigeria, comparable proportions of male and female respondents had depressive symptoms [30]. This finding is however at variance with that of a study in India where the female gender was significantly associated with depression [22]. Similarly, from the results of a study in Nepal, the mean total PHQ 9 score was significantly higher among the female respondents when compared with the males [29]. Also, a study in Bangladesh revealed that the female gender was an underlying issue for poor mental health [26]. On mental wellbeing, a significantly higher proportion of male respondents in this study had good mental wellbeing when compared with the female respondents. This was similar to the result of the study in Nepal where the mean total WEMWBS score was significantly higher among the male respondents [29]. Moreover, there has been a postulation that holistic and gender specific approaches should be taken into consideration in designing interventions towards improving the mental wellbeing of persons affected by leprosy [31].

There was a strong positive correlation between total depression and anxiety scores of the entire respondents and this was found to be statistically significant. In a study involving patients with skin diseases, there was a weak positive correlation between total depression and anxiety scores and this was also found to be statistically significant [32]. In this study also, there was a weak negative correlation between mean mental well-being score and total depression score and this finding was found to be statistically significant. From the results of a study in southern Nepal there was a strong negative correlation between total well-being score and total depression score and this finding was also found to be statistically significant [29]. These findings attest that as total depression score increases total anxiety score also increases. Conversely, as the total well-being score increases, total depression score decreases. These findings necessitate the need for the promotion of mental well-being among persons affected by leprosy or BU.

From the results of this study, respondents who had no formal education and those who had primary education had increased odds of having poor mental health/well-being when compared with those who had secondary education and above. A study in India demonstrated that low education was associated with generalized anxiety disorders among persons affected by leprosy [22]. In another study in Philippines, lower educational level increased the odds of anxiety among persons affected by leprosy [33]. Also, the respondents who were unemployed had increased odds of having poor mental health/well-being when compared with those who were employed. In a study among persons affected by leprosy attending out-patient clinics in Kaduna state, Nigeria, unemployment predicted the risk of generalized anxiety disorder [34].

From the results of this study, poor and moderate social support also increased the odds of having poor mental health/well-being when compared with those that had strong social support. A study to examine the role of perceived social support in relation to depression among individuals undergoing social isolation during the COVID-19 pandemic revealed that the risk of high levels of depressive symptoms was 63% lower among individuals who had higher levels of social support when compared with those who had low levels of social support [35]. Equally

an important finding is that individuals who were affected by leprosy were five times less likely to have poor mental health/well-being when compared to those who were affected by BU. NTDs have been identified as major drivers of mental ill health in persons affected by the diseases [6]. Perhaps, research focus has been more on leprosy when compared to BU. However, persons affected by BU have before now requested that psychological care should be part of their health care needs [8]. This also calls for more attention for BU patients in the area of mental health and well-being.

This study took place after the COVID-19 pandemic and bearing in mind the impact of the pandemic on mental health, it could have affected the results of this study. This is of essence bearing in mind the high proportion of respondents in this study that had depressive and anxiety disorders and also the proportion of respondents with poor mental wellbeing. This notwithstanding, it is important to note that previous studies have revealed that persons affected by leprosy or BU have a higher burden of poor mental health when compared with other community members [7, 9]. Also, being affected by an NTD has been identified as a risk factor for poor mental health [11]. Also, the LGAs included in the study were purposively selected based on number of notified cases between 2014 and 2018. This could affect the generalization of study findings however the results obtained are not too different from that from other parts of the country. In all, because a probability sampling technique was not used in the selection of the respondents, there could be a possibility of a selection bias since there is no evidence that all persons affected by leprosy or BU in the selected clusters were included in the study.

## Conclusion

More than half of the respondents, 57.2% had poor mental health/wellbeing meaning that the burden among persons affected by leprosy or BU is very high. There is the need to include mental health interventions in the management of persons affected with leprosy or BU. Equally important is finding a feasible, cost-effective and sustainable approach to delivering mental health care for persons affected with leprosy or BU at the community level. Improving educational status and social support of persons affected by leprosy or BU are essential. Engaging them in productive activities will be of essence.

## Supporting information

**S1 File.**
(XLSX)

## Acknowledgments

We remain grateful to persons affected by leprosy or BU for their participation in the research.

## Author Contributions

**Conceptualization:** Edmund Ndudi Ossai, Ngozi Ekeke, Amaka Esmai-Onyima, Chinwe Eze.

**Data curation:** Edmund Ndudi Ossai, Ngozi Ekeke, Amaka Esmai-Onyima, Chinwe Eze.

**Formal analysis:** Edmund Ndudi Ossai.

**Funding acquisition:** Edmund Ndudi Ossai, Ngozi Ekeke, Amaka Esmai-Onyima.

**Investigation:** Ngozi Ekeke, Amaka Esmai-Onyima, Chinwe Eze, Francis Chinawa, Obiora Iteke, Precious Henry, Ngozi Murphy-Okpala, Martin Njoku, Chukwuma Anyaike.

**Methodology:** Edmund Ndudi Ossai, Ngozi Ekeke, Joseph N. Chukwu, Charles Nwafor, Ngozi Murphy-Okpala, Martin Njoku, Anthony O. Meka.

**Project administration:** Ngozi Ekeke, Amaka Esmai-Onyima, Chinwe Eze, Precious Henry, Joseph N. Chukwu, Charles Nwafor, Ngozi Murphy-Okpala.

**Resources:** Chinwe Eze, Francis Chinawa, Obiora Iteke, Precious Henry, Joseph N. Chukwu, Charles Nwafor, Ngozi Murphy-Okpala, Martin Njoku, Anthony O. Meka, Chukwuma Anyaike.

**Supervision:** Edmund Ndudi Ossai, Ngozi Ekeke, Amaka Esmai-Onyima, Chinwe Eze, Francis Chinawa, Obiora Iteke, Precious Henry, Joseph N. Chukwu, Charles Nwafor, Martin Njoku, Anthony O. Meka, Chukwuma Anyaike.

**Validation:** Ngozi Ekeke, Francis Chinawa, Obiora Iteke, Joseph N. Chukwu, Charles Nwafor, Ngozi Murphy-Okpala, Martin Njoku, Anthony O. Meka.

**Visualization:** Precious Henry, Joseph N. Chukwu, Anthony O. Meka, Chukwuma Anyaike.

**Writing – original draft:** Edmund Ndudi Ossai.

**Writing – review & editing:** Edmund Ndudi Ossai, Ngozi Ekeke, Amaka Esmai-Onyima, Chinwe Eze, Francis Chinawa, Obiora Iteke, Precious Henry, Joseph N. Chukwu, Charles Nwafor, Ngozi Murphy-Okpala, Martin Njoku, Anthony O. Meka, Chukwuma Anyaike.

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
