## [Decision Letter · Decision Letter 0]

30 Nov 2023

PONE-D-23-33588Understanding the burden of poor mental health and wellbeing among Persons affected by Leprosy or Buruli ulcer in Nigeria: A community based cross-sectional studyPLOS ONE

Dear Dr. Ossai,

Thank you for submitting your manuscript to PLOS ONE. After careful consideration, we feel that it has merit but does not fully meet PLOS ONE’s publication criteria as it currently stands. Therefore, we invite you to submit a revised version of the manuscript that addresses the points raised during the review process.

**ACADEMIC EDITOR: **

Title

Where exactly in southern Nigeria were these studies conducted?

This should be reflected in the title as well as the methodology. For the methodology, state the specific locations were participants were recruited, which communities? And under which Local Government Areas and States? This needs to be extrapolated for clarity. State the names of the communities, LGAs and States involved in the study.

Introduction

I suggest that you start the introduction from the global effect of mental health regarding leprosy and Buruli ulcer, followed by the continental aspect and in Nigeria and finally the study area.

Methodology

Study population; line 130-133- clearly state the inclusion criteria and exclusion criteria.

Were there any chances of introduction of selection bias in your studies? You made mention that you used purposive sampling in selection of study sites. Did your findings fully reflect the entire Nigerian situation? How were you able to reduce bias in your studies? How were you able to deal with issue of confounders to ensure that your result reflects what you intend to achieve. Please, categorically state it in your methodology.

RESULTS

Please label all the tables properly, eg Table 1: Socio-demographic characteristics of the respondents (n=635)  ------(location) in Nigeria

Discussion

Limitations of the study should be the last paragraph of your discussion. Please delete the heading “limitations”,

Conclusion

Be precise by stating in percentage the burden of poor mental health/wellbeing among persons affected by leprosy or Buruli ulcer in your study.

We look forward to receiving your revised manuscript.

Kind regards,

Ayi Vandi Kwaghe, D.V.M., M.V.Sc., P.G.D.E. Ph.D., MPH

Academic Editor

PLOS ONE

Journal Requirements:

"We acknowledge the support and funding of this project by Leprosy Research Initiative. (Reference Number, 708.20.15/LRI)." ext-link-type="uri" xlink:type="simple">https://leprosyresearch.org/"

"NE, ENO and AE received the grant.

(Reference Number, 708.20.15/LRI).

Leprosy Research Initiative (LRI)

https://leprosyresearch.org/

The funders played no role in the design of the study, data collection and analysis, decision to publish and in the preparation of the manuscript."

Reviewers' comments:

Reviewer's Responses to Questions

**Comments to the Author**

1. Is the manuscript technically sound, and do the data support the conclusions?

Reviewer #1: Partly

Reviewer #2: Yes

2. Has the statistical analysis been performed appropriately and rigorously? 

Reviewer #1: I Don't Know

Reviewer #2: Yes

3. Have the authors made all data underlying the findings in their manuscript fully available?

Reviewer #1: Yes

Reviewer #2: Yes

4. Is the manuscript presented in an intelligible fashion and written in standard English?

Reviewer #1: Yes

Reviewer #2: No

5. Review Comments to the Author

Reviewer #1: 1. I would suggest using term 'Mental Health Conditions' or 'Mental Distress' instead of 'Poor Mental Health' throughout the manuscript including the title. The term 'poor mental health' would increase stigma.

2. In Introduction section- It would be good to give detailed definition of the Leprosy and BU. This is important for the audience who does not have medical background. Also, it is important to mention the prevalence of Leprosy BU in the study settings/geographical area.

3. It is not clear that why researchers chose to study prevalence of MH conditions among persons with Leprosy or BU only and not from all NTDs? The aim and objectives of this study are not clear and need to be clearly mentioned. Are researchers already involved in MH care for persons with Leprosy and BU? The way authors have mentioned need of MH intervention among persons with Leprosy is not very detailed one, and very superficial. Also, authors have mentioned effect of COVID 19 pandemic on persons with Leprosy but what about persons with BU.

4. There needs to be clarity around validity of each scale in local language and culture. Also, it would be better to mention Cronbach's Alpha (Coefficient) for mentioned scales. Please mention validity for Oslo Social Support Scale.

5. Similarly, inclusion and exclusion criteria for this study need to be mentioned.

6. Please mention training details for data collectors.

7. For consent process- please mention details of consent procedure for persons who cannot read or write. Whether authors followed ethical guidelines for this group?

8. The manuscript will need through proofread for the language. Line no 76- need to say - infections those lead to a disability instead the mentioned term. Line 98- it needs to be 'quarantine' and not 'quarantining'.

9. Also, it would be good to mention if Authors will use this data to push any policy change at local or state level or wanted to further use this data to design community-based MH intervention for persons with Leprosy and BU.

10. Is this the first study from Nigeria / Africa measuring prevalence of MH conditions among person with Leprosy and BU? If yes, then this need to be mentioned clearly.

Reviewer #2: 1. Please ensure that, before resubmission, the manuscript is carefully checked for English language and grammatical errors. It is important that the message being conveyed in the manuscript is as unambiguous as possible.

2. Lacks novelty

3. Introduction can be shortened to make it more clear

4. THe discussion is to long and it lakes coherence, try to make it short and precise

5. in the study area try to show us your study area with regard to the number of hospitals with its capacity to give service particulary poor mental health

6. in your background section try to illustrate the over all burden of poor mental health among the specified problems and clearly state the gaps and how you are going to fill it.

7. please ommit the sentence, Nigeria is the popular countary,

8. try to delet the word skin from skin tropical neglected..... start with tropical neglecte.

9. you selecte the sample purposively, so the result could not generalize the total population. pls delete sample size determination heading.

10.rather than saing study instrument, it is better to say data collection tools and try to merge it with data collection procedures.

11. the chrosectional nature of the study and purposively selection of the sample is also the limitation of the study. please include this

6. PLOS authors have the option to publish the peer review history of their article (what does this mean?). If published, this will include your full peer review and any attached files.

Reviewer #1: No

Reviewer #2: **Yes: **Assefa Agegnehu Teshome

---

## [Author Response · Author response to Decision Letter 0]

6 Dec 2023

• Drafts1.1K

Reviewer’s comments and responses by Authors

Academic Editor,

TGGHJJKCVXVXBZXNZNZNZNZNZTTTQQQRRRRR

Reviewer’s comment

Title

Where exactly in southern Nigeria were these studies conducted?

This should be reflected in the title as well as the methodology. For the methodology, state the specific locations were participants were recruited, which communities? And under which Local Government Areas and States? This needs to be extrapolated for clarity. State the names of the communities, LGAs and States involved in the study.

Author response

Many thanks for this comment. The states and local government areas have been included. The communities may be very difficult to define. This is because the NTBLCP which is the agency of Government in charge of leprosy and Burruli ulcer activities are structured along the three tiers of government in Nigeria which are the Federal, State and Local Government Areas. 

Reviewer’s comment

Introduction

I suggest that you start the introduction from the global effect of mental health regarding leprosy and Buruli ulcer, followed by the continental aspect and in Nigeria and finally the study area.

Author response

This has been included. See lines 106-108.

Methodology

Reviewer’s comment

Study population; line 130-133- clearly state the inclusion criteria and exclusion criteria.

Were there any chances of introduction of selection bias in your studies? You made mention that you used purposive sampling in selection of study sites. Did your findings fully reflect the entire Nigerian situation? How were you able to reduce bias in your studies? How were you able to deal with issue of confounders to ensure that your result reflects what you intend to achieve. Please, categorically state it in your methodology.

Author response

The inclusion and exclusion criteria have been included. See study population. The officials of NTBLCP were fully involved in the study and they have a register of all persons diagnosed with leprosy or BU in all the LGAs of the country. So selection bias was not a problem in the study. The LGAs were purposively selected based on case notifications between 2014 and 2018. This was purely to ensure maximal impact and this may not have affected the results of the study

Reviewer’s comment

RESULTS

Please label all the tables properly, eg Table 1: Socio-demographic characteristics of the respondents (n=635) ------(location) in Nigeria

Author response

This is noted. Corrected.

Reviewer’s comment

Discussion

Limitations of the study should be the last paragraph of your discussion. Please delete the heading “limitations”,

Author response

The heading ‘limitations’ has been deleted.

Reviewer’s comment

Conclusion

Be precise by stating in percentage the burden of poor mental health/wellbeing among persons affected by leprosy or Buruli ulcer in your study.

Author response

Included

Reviewer’s Comment

Reviewer #1: 1. I would suggest using term 'Mental Health Conditions' or 'Mental Distress' instead of 'Poor Mental Health' throughout the manuscript including the title. The term 'poor mental health' would increase stigma.

Author response

This suggestion is appreciated. However, the outcome variable in this study is poor mental health/wellbeing. This is because the variable was derived from the use of three validated questionnaires including PHQ9, GAD7 and WEMWBS. That is why it appears that poor mental health/wellbeing may be the most appropriate to represent this variable. 

Reviewer’s Comment

2. In Introduction section- It would be good to give detailed definition of the Leprosy and BU. This is important for the audience who does not have medical background. Also, it is important to mention the prevalence of Leprosy BU in the study settings/geographical area.

Author response

This has been included, see lines 74-76.

Reviewer’s Comment

3. It is not clear that why researchers chose to study prevalence of MH conditions among persons with Leprosy or BU only and not from all NTDs? The aim and objectives of this study are not clear and need to be clearly mentioned. Are researchers already involved in MH care for persons with Leprosy and BU? The way authors have mentioned need of MH intervention among persons with Leprosy is not very detailed one, and very superficial. Also, authors have mentioned effect of COVID 19 pandemic on persons with Leprosy but what about persons with BU.

Author response

This study was focused on persons affected by leprosy or Buruli ulcer. In-fact, the study prompted another one which included other NTDs. Objective: This study was designed to assess the mental health and wellbeing of persons affected by leprosy or BU in Nigeria. In Nigeria, mental health services are not included in the health services for persons affected by leprosy or BU and the finding of this study should suggest the need for such services. The literature has much on the effect of COVID-19 pandemic on persons affected by leprosy unlike BU.

Reviewer’s Comment

4. There needs to be clarity around validity of each scale in local language and culture. Also, it would be better to mention Cronbach's Alpha (Coefficient) for mentioned scales. Please mention validity for Oslo Social Support Scale.

Author response

We did not include validity for Oslo Social Support Scale in our study but the tool has been used several times in our study setting.

Reviewer’s Comment

5. Similarly, inclusion and exclusion criteria for this study need to be mentioned.

Author response

The inclusion and exclusion criteria of the study have been included. See study population.

Reviewer’s Comment

6. Please mention training details for data collectors.

Author response

Training of research assistants centered on the use of the study tools, obtaining informed consent and the need for confidentiality and also privacy when collecting information from the respondents, etc. Officials of NTBLCP who accompanied the research assistants identified persons affected by leprosy or BU for interview.

Reviewer’s Comment

7. For consent process- please mention details of consent procedure for persons who cannot read or write. Whether authors followed ethical guidelines for this group?

Author response

The process of obtaining informed consent took place using the local language so an interpreter was present and also a witness. When the process was over, the respondents will explain all that they understood from the process. It is only when this was judged as adequate that the responded thumb printed on the informed consent form followed by the signature of the witness.

Reviewer’s Comment

8. The manuscript will need through proofread for the language. Line no 76- need to say - infections those lead to a disability instead the mentioned term. Line 98- it needs to be 'quarantine' and not 'quarantining'.

Author response

This has been taken care of. Many thanks.

Reviewer’s Comment

9. Also, it would be good to mention if Authors will use this data to push any policy change at local or state level or wanted to further use this data to design community-based MH intervention for persons with Leprosy and BU.

Author response

The reviewer is right. A community based MH intervention for persons affected by leprosy or BU is on-going.

Reviewer’s Comment

10. Is this the first study from Nigeria / Africa measuring prevalence of MH conditions among person with Leprosy and BU? If yes, then this need to be mentioned clearly.

Author response

This is not the first study from Nigeria that measured the prevalence of MH conditions among persons affected by leprosy or BU. However, it could be the first that involved such a high number, (n=635) and spread. The study took place across six states in two geo-political zones of the country.

Reviewer’s Comment

Reviewer #2: 1. Please ensure that, before resubmission, the manuscript is carefully checked for English language and grammatical errors. It is important that the message being conveyed in the manuscript is as unambiguous as possible.

Author response

Many Thanks for the comment. Grammatical errors noticed have been corrected.

Reviewer’s Comment

3. Introduction can be shortened to make it more clear

Author response

Thanks. Information is noted.

Reviewer’s Comment

4. The discussion is to long and it lakes coherence, try to make it short and precise

Author response

Noted. Done,

Reviewer’s Comment

5. in the study area try to show us your study area with regard to the number of hospitals with its capacity to give service particulary poor mental health

Author response

The observation that hospitals with such capacity are few and concentrated on urban areas which are very far from these communities prompted the MH intervention that is ongoing which is community based by making use of available human resources who are trained for that purpose.

Reviewer’s Comment

6. in your background section try to illustrate the over all burden of poor mental health among the specified problems and clearly state the gaps and how you are going to fill it.

Author response

See lines 90-97 and lines 93-112.

Reviewer’s Comment

7. please ommit the sentence, Nigeria is the popular countary,

Author response

We indicated that, ‘Nigeria is the most populous country in Africa,’ and that is a fact.

Reviewer’s Comment

8. try to delet the word skin from skin tropical neglected..... start with tropical neglecte.

Author response

Corrected.

Reviewer’s Comment

9. you selecte the sample purposively, so the result could not generalize the total population. pls delete sample size determination heading.

Author response

NTBLCP is the agency of Government in Nigeria responsible for TB, leprosy and BU activities in Nigeria. We used their data to select ten LGAs in southern Nigeria that reported the highest cases of leprosy and BU between 2014 and 2018. These ten LGAs were then purposively selected for maximum impact. The heading ‘Sample size determination’ is important because there we explained that the study was a total population study of all persons affected by leprosy or BU in the selected LGAs. 

Reviewer’s Comment

10.rather than saing study instrument, it is better to say data collection tools and try to merge it with data collection procedures.

Author response

Correction done.

Reviewer’s Comment

11. the chrosectional nature of the study and purposively selection of the sample is also the limitation of the study. please include this

Author response

Included. Thanks

---

## [Decision Letter · Decision Letter 1]

14 Feb 2024

PONE-D-23-33588R1Understanding the burden of poor mental health and wellbeing among Persons affected by Leprosy or Buruli ulcer in Nigeria: A community based cross-sectional studyPLOS ONE

Dear Dr. Ossai, 

Thank you for submitting your manuscript to PLOS ONE. After careful consideration, we feel that it has merit but does not fully meet PLOS ONE’s publication criteria as it currently stands. Therefore, we invite you to submit a revised version of the manuscript that addresses the points raised during the review process.

**ACADEMIC EDITOR:**

Introduction

Authors should please list the studies that dealt with mental health and collectively elaborate on the findings of the studies to limit the bulk of the introduction and improve the clarity of their manuscript. The introduction needs to be fine-tuned properly for better comprehension of the available information. Authors still need to work on information flow.

Methods

It is a fact that Nigeria is the most populous nation in Africa but what we need under the study setting is for you to state and cite the country’s population for readers to have an idea of the population.

Study population

Clearly state the inclusion and exclusion criteria for instance “the inclusion criteria for the study were persons affected with leprosy or BU who were registered…………………………………………while the exclusion criteria for our study were……………………………………”.

Line 147: you made mention that age was not a barrier to inclusion in your study, why did you put age as part of the inclusion criteria in your initial statement? Please, delete age in your inclusion criteria if it was not used in the selection of participants. Be precise in your statements.

Line 152 Sample size determination and sampling technique

Please delete determination of sample size as indicated by earlier reviewer

Under sample technique, you can state the sampling method and the method of recruitment of participants in the study. It is also good to state the total number of participants that participated in the study.

The authors made mention of limited bias in the study due to the fact that areas with high risk of leprosy and buruli ulcer were selected for the study to improve the quality of data that will be obtained. Also, they made mention of the number of study participants. This should be clearly stated under the method and not just in the response to the reviewers bearing in mind that there could be possibility of bias

 even with large samples since the participants were not randomly sampled.

Line 159 Data collection tools/methods please clearly state the website of the data collection tools used in the study if available

Line 200 please write the heading in full. You can abbreviate in bracket

Line 239-241; the duration of the study should not be stated under ethical clearance. It should be stated under the study design.

**Results                                      **

Please ensure that the total number of participants/respondents is indicated in all the titles of the tables presented. Example; Assessment of anxiety symptoms among persons (n=635) affected by leprosy or Buruli ulcer in Nigeria using GAD7. You can delete the n=635 within the tables.

Line 301; χ^2^=0,010 please replace the comma with a dot in stating the Chi square value

Line 304, Table 7, please appropriately place the 1 value for the GAD 7 score compared with the GAD 7 score

Discussion

Line 342 From the results of this study also? Why not write “the study also reveals…..”. Please endeavor to edit the entire manuscript properly.

The discussion should follow the pattern to which your results were presented. Your discussion need to be concise and articulated to ensure the flow of the discussion. Eg studies in India have indicated the range of depressive disorders from 30-53% (--citations). I suggest that repetition of points should be avoided, for example Line 359-360, 433-434, 446-447; all referring NTDs as a risk factor in developing poor mental health. Regarding the possible effect of COVID-19 accruing to the high level of depression, anxity and poor mental health in this study can be presented in a concise manner.

Line 447-450 should be the limitation of the study. It should be the last paragraph of the discussion. Purposive sampling is a limitation based on sampling method. Preferably, probability sampling should be used for the generalization of results and to eliminate bias. Could it be that those areas that were not sampled, if sampled, the results obtained could have differed based on sociocultural and economic differences? The sample size of the study was large which might have aided in the research but may not completely rule out elements of bias. Other similar studies that were cited, did they use probability sampling? The issue of selection bias may not be totally ruled out due to the nature of the sampling involved. I suggest you mention it as part of the limitation of your study.

Conclusion

Line 453 Please rephrase, do not start a sentence with “with”.

We look forward to receiving your revised manuscript.

Kind regards,

Ayi Vandi Kwaghe, D.V.M., M.V.Sc., P.G.D.E. Ph.D., MPH, FETP

Academic Editor

PLOS ONE

Journal Requirements:

Reviewers' comments:

Reviewer's Responses to Questions

**Comments to the Author**

1. If the authors have adequately addressed your comments raised in a previous round of review and you feel that this manuscript is now acceptable for publication, you may indicate that here to bypass the “Comments to the Author” section, enter your conflict of interest statement in the “Confidential to Editor” section, and submit your "Accept" recommendation.

Reviewer #3: (No Response)

Reviewer #4: (No Response)

2. Is the manuscript technically sound, and do the data support the conclusions?

Reviewer #3: Partly

Reviewer #4: No

3. Has the statistical analysis been performed appropriately and rigorously? 

Reviewer #3: Yes

Reviewer #4: (No Response)

4. Have the authors made all data underlying the findings in their manuscript fully available?

Reviewer #3: Yes

Reviewer #4: (No Response)

5. Is the manuscript presented in an intelligible fashion and written in standard English?

Reviewer #3: Yes

Reviewer #4: No

6. Review Comments to the Author

Reviewer #3: I think this is an interesting and useful study. However, I believe it needs some modifications for publication.

It is recommended that the number of participants be added to the abstract.

We recommend adding information such as sensitivity, specificity, etc. regarding the tools used.

Please elaborate on why you believe selection bias is low. Please also reflect this in your manuscript.

We recommend adding suggestions for future research.

Please review your manuscript carefully. (e.g. line 329)

Reviewer #4: Reviewer # Comment

Dear authors, the topic of your study has been registered as a study protocol in which authors included except some of them.

STUDY PROTOCOL

A Cluster Randomized Trial for Improving Mental Health and Well-Being of Persons Affected by Leprosy or Buruli Ulcer in Nigeria

A Study Protocol

Ekeke, Ngozi1; Ossai, Edmund Ndudi2,; Kreibich, Saskia3; Onyima, Amaka4; Chukwu, Joseph1; Nwafor, Charles1; Meka, Anthony1; Murphy-Okpala, Ngozi1; Henry, Precious1; Eze, Chinwe1

link: https://journals.lww.com/ijmy/fulltext/2022/11020/a_cluster_randomized_trial_for_improving_mental.1.aspx

Therefore, objectives are mentioned in the study protocol, I expect they will be addressed and published (I do not know when). What your study adds different from the planed study? It seems that authors need to publish the baseline information of patients. I think this is not recommended as it may be considered fragmenting one paper to multiple separate paper for the matter of publication. It is better to conduct and publish the trial timely. In my view the present manuscript should be rejected due to the points raised above.

7. PLOS authors have the option to publish the peer review history of their article (what does this mean?). If published, this will include your full peer review and any attached files.

Reviewer #3: No

Reviewer #4: No

---

## [Author Response · Author response to Decision Letter 1]

25 Feb 2024

Reviewer’s comments and response from Authors

Introduction

Authors should please list the studies that dealt with mental health and collectively elaborate on the findings of the studies to limit the bulk of the introduction and improve the clarity of their manuscript. The introduction needs to be fine-tuned properly for better comprehension of the available information. Authors still need to work on information flow.

Author response

Thanks for the comment. Very few studies focused on mental health of persons affected by leprosy or BU in Nigeria. Those found were included. Studies from other parts of the world especially India were cited in great numbers in the discussion. Authors however worked on the information flow in the Introduction section. I hope the manuscript is acceptable in this present form.

Reviewer’s comment

Methods

It is a fact that Nigeria is the most populous nation in Africa but what we need under the study setting is for you to state and cite the country’s population for readers to have an idea of the population.

Author response

This has been included. See study setting.

Reviewer’s comment

Study population

Clearly state the inclusion and exclusion criteria for instance “the inclusion criteria for the study were persons affected with leprosy or BU who were registered…………………………………………while the exclusion criteria for our study were……………………………………”.

Line 147: you made mention that age was not a barrier to inclusion in your study, why did you put age as part of the inclusion criteria in your initial statement? Please, delete age in your inclusion criteria if it was not used in the selection of participants. Be precise in your statements.

Author response

First the individuals affected by leprosy or Buruli ulcer must have been registered for treatment under the care of NTBLCP which is the agency of Government responsible for the management of tuberculosis, leprosy and Buruli ulcer in the country. After this, age was applied differently for the inclusion of persons affected by leprosy or Buruli ulcer in the study.

‘For persons affected by leprosy only those who were 18 years and above were included in the study. However, because of the epidemiology of BU in the country, age was not a barrier for inclusion of persons affected by BU in the study.’

Reviewer’s comment

Line 152 Sample size determination and sampling technique

Please delete determination of sample size as indicated by earlier reviewer

Under sample technique, you can state the sampling method and the method of recruitment of participants in the study. It is also good to state the total number of participants that participated in the study.

Author response

This has been included. See sampling technique.

Reviewer’s comment

The authors made mention of limited bias in the study due to the fact that areas with high risk of leprosy and buruli ulcer were selected for the study to improve the quality of data that will be obtained. Also, they made mention of the number of study participants. This should be clearly stated under the method and not just in the response to the reviewers bearing in mind that there could be possibility of bias even with large samples since the participants were not randomly sampled.

Author response

Thanks for the observation.

The limitations of the study have been improved upon. 

Reviewer’s comment

Line 159 Data collection tools/methods please clearly state the website of the data collection tools used in the study if available

Author response

Included. Thanks

Reviewer’s comment

Line 200 please write the heading in full. You can abbreviate in bracket

Author response

This has been done. My appreciations.

Reviewer’s comment

Line 239-241; the duration of the study should not be stated under ethical clearance. It should be stated under the study design.

Author response

Included.

Reviewer’s comment

Results 

Please ensure that the total number of participants/respondents is indicated in all the titles of the tables presented. Example; Assessment of anxiety symptoms among persons (n=635) affected by leprosy or Buruli ulcer in Nigeria using GAD7. You can delete the n=635 within the tables.

Author response

Done

Reviewer’s comment

Line 301; χ2=0,010 please replace the comma with a dot in stating the Chi square value

Author response

Corrected. Thanks.

Reviewer’s comment

Line 304, Table 7, please appropriately place the 1 value for the GAD 7 score compared with the GAD 7 score

Author response

Done

Reviewer’s comment

Discussion

Line 342 From the results of this study also? Why not write “the study also reveals…..”. 

Author response

Corrected.

Reviewer’s comment

Please endeavor to edit the entire manuscript properly.

Author response

I have done that. Thanks

Reviewer’s comment

The discussion should follow the pattern to which your results were presented. Your discussion need to be concise and articulated to ensure the flow of the discussion. Eg studies in India have indicated the range of depressive disorders from 30-53% (--citations). I suggest that repetition of points should be avoided, for example Line 359-360, 433-434, 446-447; all referring NTDs as a risk factor in developing poor mental health. Regarding the possible effect of COVID-19 accruing to the high level of depression, anxity and poor mental health in this study can be presented in a concise manner.

Author response

Corrected.

Reviewer’s comment

Line 447-450 should be the limitation of the study. It should be the last paragraph of the discussion. Purposive sampling is a limitation based on sampling method. Preferably, probability sampling should be used for the generalization of results and to eliminate bias. Could it be that those areas that were not sampled, if sampled, the results obtained could have differed based on sociocultural and economic differences? The sample size of the study was large which might have aided in the research but may not completely rule out elements of bias. Other similar studies that were cited, did they use probability sampling? The issue of selection bias may not be totally ruled out due to the nature of the sampling involved. I suggest you mention it as part of the limitation of your study.

Author response

The limitations of the study have been reviewed.

Thanks

Reviewer’s comment

Conclusion

Line 453 Please rephrase, do not start a sentence with “with”.

Author response

Done.

---

## [Editor Report · Decision Letter 2]

27 Feb 2024

PONE-D-23-33588R2Understanding the burden of poor mental health and wellbeing among Persons affected by Leprosy or Buruli ulcer in Nigeria: A community based cross-sectional studyPLOS ONE

Dear Dr. Ossai,

Thank you for submitting your manuscript to PLOS ONE. After careful consideration, we feel that it has merit but does not fully meet PLOS ONE’s publication criteria as it currently stands. Therefore, we invite you to submit a revised version of the manuscript that addresses the points raised during the review process.

**ACADEMIC EDITOR: **

The reviewer below raised critical questions which the authors did not respond to. I suggest the authors respond to the questions raised by the reviewer as stated below:

Reviewer # Comment

Dear authors, the topic of your study has been registered as a study protocol in which authors included except some of them.

STUDY PROTOCOL

A Cluster Randomized Trial for Improving Mental Health and Well-Being of Persons Affected by Leprosy or Buruli Ulcer in Nigeria

A Study Protocol

Ekeke, Ngozi1; Ossai, Edmund Ndudi2,; Kreibich, Saskia3; Onyima, Amaka4; Chukwu, Joseph1; Nwafor, Charles1; Meka, Anthony1; Murphy-Okpala, Ngozi1; Henry, Precious1; Eze, Chinwe1

link: https://journals.lww.com/ijmy/fulltext/2022/11020/a_cluster_randomized_trial_for_improving_mental.1.aspx

Therefore, objectives are mentioned in the study protocol, I expect they will be addressed and published (I do not know when). What your study adds different from the planed study? It seems that authors need to publish the baseline information of patients. I think this is not recommended as it may be considered fragmenting one paper to multiple separate paper for the matter of publication. It is better to conduct and publish the trial timely. In my view the present manuscript should be rejected due to the points raised above.

We look forward to receiving your revised manuscript.

Kind regards,

Ayi Vandi Kwaghe, D.V.M., M.V.Sc., P.G.D.E. Ph.D., MPH, FETP

Academic Editor

PLOS ONE
---

## [Author Response · Author response to Decision Letter 2]

2 Mar 2024

Reviewer comment and response from Authors

Reviewer # Comment

Dear authors, the topic of your study has been registered as a study protocol in which authors included except some of them.

STUDY PROTOCOL

A Cluster Randomized Trial for Improving Mental Health and Well-Being of Persons Affected by Leprosy or Buruli Ulcer in Nigeria

A Study Protocol

Ekeke, Ngozi1; Ossai, Edmund Ndudi2,; Kreibich, Saskia3; Onyima, Amaka4; Chukwu, Joseph1; Nwafor, Charles1; Meka, Anthony1; Murphy-Okpala, Ngozi1; Henry, Precious1; Eze, Chinwe1

link: https://journals.lww.com/ijmy/fulltext/2022/11020/a_cluster_randomized_trial_for_improving_mental.1.aspx

Therefore, objectives are mentioned in the study protocol, I expect they will be addressed and published (I do not know when). What your study adds different from the planed study? It seems that authors need to publish the baseline information of patients. I think this is not recommended as it may be considered fragmenting one paper to multiple separate paper for the matter of publication. It is better to conduct and publish the trial timely. In my view the present manuscript should be rejected due to the points raised above.

Author response 

Dear Editor,

Thank you for the comment. The Reviewer is right and indeed the Authors of the manuscript did not hide this fact during the submission process. This was when we were asked if the manuscript has been submitted to any of the PLOS Journals. We admitted that when we submitted the baseline result to PLOS NTD for publication, the Editor and Reviewers advised and guided the Authors to publish the result as a cross-sectional study which we eventually adhered to. 

We have also observed that with the comments of the Reviewers from PLOS NTD, the presentation of the results of this study as submitted is very different from what was included in the study protocol. The outcome variable of this study as presented is ‘poor mental health/wellbeing’ which is entirely a new concept. Another observation is that there are very few studies in Nigeria that focused on the mental health and wellbeing of persons affected by leprosy or BU and none again has as much sample size and spread (five states and two geo-political zones in Nigeria).To a large extent, this study as presented brought to the fore the huge burden of poor mental health/wellbeing among persons affected by leprosy or BU in Nigeria. The focus of the result as contained in the study protocol will be on the effect of the intervention.

All these points as noted above is not to justify any wrong doing on the part of the Authors but to plead with the Editor and Reviewers of PLOS ONE Journal to see the contribution of this study to the body of knowledge especially as it pertains to Nigeria and approve that the peer review of this manuscript should continue.

---

## [Editor Report · Decision Letter 3]

20 May 2024

Understanding the burden of poor mental health and wellbeing among Persons affected by Leprosy or Buruli ulcer in Nigeria: A community based cross-sectional study

PONE-D-23-33588R3

Dear Dr. Ossai,

We’re pleased to inform you that your manuscript has been judged scientifically suitable for publication and will be formally accepted for publication once it meets all outstanding technical requirements.

Kind regards,

Ayi Vandi Kwaghe, D.V.M., M.V.Sc., P.G.D.E. Ph.D., MPH, FETP

Academic Editor

PLOS ONE
---

## [Editor Report · Acceptance letter]

23 May 2024

PONE-D-23-33588R3 

PLOS ONE

Dear Dr. Ossai, 

I'm pleased to inform you that your manuscript has been deemed suitable for publication in PLOS ONE. Congratulations! Your manuscript is now being handed over to our production team.

Kind regards, 

on behalf of

Dr. Ayi Vandi Kwaghe 

Academic Editor

PLOS ONE